# Genetic Basis of the Epidemiological Features and Clinical Significance of Renal Hypouricemia

**DOI:** 10.3390/biomedicines10071696

**Published:** 2022-07-13

**Authors:** Masayuki Hakoda, Kimiyoshi Ichida

**Affiliations:** 1Department of Nutritional Sciences, Faculty of Human Ecology, Yasuda Women’s University, 6-13-1 Yasuhigashi, Asaminami-ku, Hiroshima 731-0153, Japan; 2Department of Pathophysiology, Tokyo University of Pharmacy and Life Sciences, 1432-1 Horinouchi, Tokyo 192-0392, Japan; ichida@toyaku.ac.jp

**Keywords:** renal hypouricemia, serum uric acid, epidemiology, urate transporter 1, mutation

## Abstract

A genetic defect in urate transporter 1 (URAT1) is the major cause of renal hypouricemia (RHUC). Although RHUC is detected using a serum uric acid (UA) concentration <2.0 mg/dL, the relationship between the genetic state of *URAT1* and serum UA concentration is not clear. Homozygosity and compound heterozygosity with respect to mutant *URAT1* alleles are associated with a serum UA concentration of <1.0 mg/dL and are present at a prevalence of ~0.1% in Japan. In heterozygous individuals, the prevalence of a serum UA of 1.1–2.0 mg/dL is much higher in women than in men. The frequency of mutant *URAT1* alleles is as high as 3% in the general Japanese population. The expansion of a specific mutant *URAT1* allele derived from a single mutant gene that occurred in ancient times is reflected in modern Japan at a high frequency. Similar findings were reported in Roma populations in Europe. These phenomena are thought to reflect the ancient migration history of each ethnic group (founder effects). Exercise-induced acute kidney injury (EI-AKI) is mostly observed in individuals with homozygous/compound heterozygous *URAT1* mutation, and laboratory experiments suggested that a high UA load on the renal tubules is a plausible mechanism for EI-AKI.

## 1. Introduction

Uric acid (UA) is an end product of purine metabolic pathways and is poorly soluble in body fluids [1]. This characteristic causes the accumulation of UA in patients with hyperuricemia, causing gout, urinary calculi, and renal damage [1]. UA was also implicated in the development of cardiovascular diseases in some ways [2], the scavenging of reactive oxygen species [3], and as an endogenous danger signal, alerting the immune system to the presence of dying cells [4]. However, the clinical significance of hypouricemia is unclear, except with respect to exercise-induced acute kidney injury (EI-AKI) and renal calculi [5]. The former occurs in patients with renal hypouricemia (RHUC), which is caused by reduced reabsorption of UA by the proximal tubules, resulting in greater excretion of UA in the urine [5], while the latter occurs in the presence of hypouricemia owing to RHUC or xanthinuria. Xanthinuria is much less common, and for most patients with hypouricemia in the general population, RHUC is the etiology [5]. Since the identification of urate transporter 1 (URAT1) [6] and glucose transporter 9 (GLUT9) [7], our understanding of the epidemiology and pathophysiology of RHUC has substantially improved.

For example, the prevalence of RHUC is different between men and women when the definition of hypouricemia is set to be a serum UA (SUA) concentration of ≤2.0 mg/dL, and this difference can be explained by different levels of SUA concentration in individuals that are heterozygous for mutant *URAT1* alleles between men and women. Further, a relatively high prevalence of RHUC in Japan is based on a high frequency of a specific mutant *URAT1* allele among the general population. This prevalent mutant *URAT1* allele was derived from a single mutant gene that occurred in ancient times and then expanded during the population history. A similar phenomenon was observed in Roma populations in Europe. The pathophysiology of EI-AKI was investigated using a new animal model with a defect of URAT1 function, where the importance of high UA load on renal tubules for the development of EI-AKI was demonstrated. The importance of a high UA load on renal tubules for the development of decreased kidney function was also demonstrated by clinical studies of potent inhibitors of URAT1. Individuals that are heterozygous for mutant *URAT1* alleles are associated with potential risks of kidney damage through a high UA load on renal tubules and are present at high frequencies among Japanese, Korean, and Romani people.

## 2. Prevalence of Hypouricemia

Hypouricemia is frequently defined using an SUA concentration of ≤2.0 mg/dL [5]. Using this reference value, the prevalence of hypouricemia in the general population was reported for Japan [8,9,10,11] and Korea [12]. These two countries have similar prevalences of hypouricemia at between 0.2 and 0.4%. The prevalence of hypouricemia is higher in women than in men in both countries, ranging from 0.4% to 0.6% in women and from 0.1% to 0.2% in men. The prevalence tends to be higher in younger women than in older women in both these countries [8,12]. However, the prevalence of hypouricemia in the general population of other countries has not been reported recently, probably because this abnormality is more common in Japan and Korea than elsewhere. Indeed, an assessment of 3200 blood donors (2097 men and 1103 women) in four studies conducted since 1962 in Germany identified only three individuals with SUA concentrations ≤2.0 mg/dL [13]. These three individuals were all women; no men with an SUA concentration <2.0 mg/dL were identified, even though a larger number of men were included.

In Japan, the prevalence of hypouricemia, with an SUA concentration ≤1.0 mg/dL, was shown to be similar for men and women at ~0.1% in a large study of >10,000 individuals of each sex recruited from the general population [10,11]. This is in contrast to the result of the studies described above, which showed that the prevalence of an SUA concentration ≤2.0 mg/dL in women is two or three times higher than that in men in Japan and Korea. In addition, in individuals with an SUA concentration ≤1.0 mg/dL, there are small but distinct groups of both men and women with SUA concentrations of 0.6–0.7 mg/dL [9,10]. This may suggest that distinct pathological mechanisms operate in individuals with hypouricemia with ≤1.0 mg/dL and 1.1–2.0 mg/dL.

## 3. Genetic Basis for the Epidemiological Features of Hypouricemia

UA is filtered through the glomeruli in the kidneys and approximately 90% of the filtered UA is reabsorbed in the renal proximal tubules. Thus, only 10% of the filtered UA is excreted in the urine. A deficiency in the renal reabsorption of UA causes a low SUA concentration (RHUC), which is characterized by a high UA-to-creatinine clearance ratio (the fractional excretion of uric acid (FEUA)). The urate transporter URAT1, located on the apical membrane of renal proximal tubular cells, and GLUT9, located on the basolateral membrane, are the principal mediators of the reabsorption of UA because homozygous or compound heterozygous deficiency in these transporters causes the SUA concentration to be low at ≤1.0 mg/dL [6,14,15], whereas that of healthy individuals is generally between 3.0 and 7.0 mg/dL.

The RHUC caused by URAT1 deficiency is called RHUC type 1 (OMIM: 220150), and more than 30 mutant *URAT1* genes that cause a functional deficiency of URAT1 were reported (HGMD^®^ 2022. 4). However, in Japan, two mutant URAT1 alleles—W258X (rs121907892) and R90H (rs121907896)—predominate [14,15,16]. A study of 1875 individuals randomly selected from among the residents of a Japanese city showed that the allele frequency of W258X was as high as 2.37% (89 alleles), that of R90H was 0.40% (15 alleles), and only one other mutant *URAT1* allele (D313A) was identified [16]. Similar prevalences of the *URAT1* W258X allele were also reported in other Japanese populations [17,18,19] (Table 1). Therefore, the total frequency of mutant *URAT1* alleles seems to be high at nearly 3% in Japan.

The genotype frequencies measured previously did not deviate from a Hardy–Weinberg equilibrium [16]. Therefore, the prevalence of the two mutant *URAT1* alleles (homozygote or compound heterozygote) can be estimated to be ~0.09% (square of the mutant allele frequency of ~3%) in Japan, which is similar to the reported prevalence of ~0.1% for individuals with an SUA concentration ≤1.0 mg/dL in the general population [10,11]. Indeed, Kawamura et al. reported that 11 of 17 men and 13 of 19 women from the general population whose SUA concentration was ≤1.0 mg/L were homozygous or compound heterozygous for the W258X and the R90H alleles [11] (Table 2), and only two with an SUA concentration ≤ 1.0 mg/dL did not have either the W258X or R90H allele (Table 2). Hamajima et al. studied a different sample of the general Japanese population and found that two out of three men and all of the three women whose SUA concentration was <1.0 mg/L were homozygous for the W258X allele [19]. Almost all of the individuals (48 out of 52) who did not have chronic renal failure and had two mutant *URAT1* alleles (homozygotes or compound heterozygotes) had SUA concentrations of <1.0 mg/dL [15]. These results suggest that most people with a low SUA concentration of ≤1.0 mg/dL are homozygous or compound heterozygous for mutant *URAT1* alleles in Japan. This implies that hypouricemia with an SUA concentration of ≤1.0 mg/dL is, in essence, a genetic disorder.

Kawamura et al. reported that the prevalence of an SUA concentration of 1.1–2.0 mg/dL is low in men (0.03%), approximately one-fifth of that of an SUA concentration of ≤1.0 mg/L [11] (Table 3). In contrast, the prevalence of an SUA concentration of 1.1–2.0 mg/dL in women (0.41%) is approximately three times higher than that of an SUA concentration of ≤1.0 mg/L (Table 3). Therefore, the prevalence of an SUA concentration of 1.1–2.0 mg/dL in women is more than 10 times higher than that of men. In this study, approximately two-thirds of women with an SUA concentration of 1.1–2.0 mg/dL were heterozygous for one of the two common mutant *URAT1* alleles (W258X and R90H) [11] (Table 2). Therefore, the higher prevalence of hypouricemia (SUA concentration ≤ 2.0 mg/dL) in women than in men can be explained by a higher prevalence of heterozygosity for a mutant *URAT1* allele, causing an SUA concentration of 1.1–2.0 mg/dL, in women than in men (Figure 1). The SUA concentrations of men who are heterozygous for a *URAT1* mutation were most frequently within the range of 3.0–4.9 mg/dL [19]. The lower SUA concentrations in women than in men who are heterozygous for a *URAT1* mutation are most probably explained by the effect of female sex hormones to increase the renal excretion of uric acid [20]. Consistent with this, a higher prevalence of hypouricemia (SUA concentration ≤ 2.0 mg/dL) was found in younger (<50 years old) than in older (≥50 years old) women in both Japan and Korea [8,12]. Women of <50 years of age are typically premenopausal, with higher concentrations of female sex hormones.

Another cause of RUHC is a genetic defect in *GLUT9*, which encodes a protein that transports reabsorbed UA from the proximal tubular cells into circulation (RHUC type 2, OMIM:612076) [21]. GLUT9 mutations seem to cause a more severe defect in UA reabsorption in the kidney than URAT1 mutations because the FEUA is sometimes >100% in patients with the former [21]. Matsuo et al. screened the genomic sequence of the *GLUT9* gene in 70 Japanese individuals with an SUA concentration ≤3.0 mg/dL and found two mutant *GLUT9* alleles (the *URAT1* W258X allele was identified in 47 out of 70 individuals) [7]. These *GLUT9* mutant alleles were not identified in 130 randomly selected healthy individuals (260 alleles). Anzai et al. identified a different mutant *GLUT9* allele (P412R) in patients with RHUC who did not have a *URAT1* mutation, but could not find the same mutation in 50 other randomly selected healthy Japanese people (100 alleles) [22]. Therefore, mutant *GLUT9* alleles appear to be rarer than mutant *URAT1* alleles in Japan given that the allele frequency of the latter was found to be close to 3% in the general population, as described above. One study of 17 patients with renal hypouricemia and exercise-induced acute kidney injury (EI-AKI), for whom both the *URAT1* and *GLUT9* genes were sequenced, showed that 13 were homozygous or compound heterozygous for mutant URAT1 alleles, two were heterozygous for mutant *URAT1* alleles, and only one was compound heterozygous for mutant *GLUT9* alleles (both of the two mutant GLUT9 alleles differed from those reported by Matsuo et al. [7] or Anzai et al. [22,23]).

Among Roma populations and in Korea, where the same *URAT1* mutant alleles were repeatedly identified, lower frequencies of *GLUT9* mutant alleles than of *URAT1* mutant alleles were reported [24,25,26]. In China, screening of the *URAT1* and *GLUT9* genes for mutations was performed in 31 individuals with hypouricemia (SUA concentration ≤2.0 mg/dL) selected from the general population [27], and this showed that the number of people with *GLUT9* mutations was lower than that of people with *URAT1* mutations (two vs. four, no mutations in the *URAT1* or *GLUT9* genes were found in the other 25 individuals). In this study, neither the *URAT1* W258X nor R90H mutant alleles, which are the predominant genetic causes of renal hypouricemia in Japan and Korea, were identified. 

## 4. Genetic Basis for the Differing Prevalence of Renal Hypouricemia between Different Ethnic Groups

As described above, the same mutations of the *URAT1* gene were identified in multiple unrelated patients with RHUC in Japan. An analysis of 13 genetic markers flanking the most prevalent *URAT1* mutation (W258X) showed that all of the mutant alleles were derived from one mutant haplotype that developed approximately 6820 years ago [15]. Because the genotype frequencies do not deviate from Hardy–Weinberg equilibrium [16], it seems unlikely that a survival advantage was conferred by the mutant *URAT1* allele. Interestingly, the same mutant *URAT1* allele (W258X) was found in studies of individuals with hypouricemia in Korea and this was also the most prevalent (57%) mutant *URAT1* allele in the Korean population [26]. Haplotype analysis of the W258X mutant allele was also performed in the Korean population [28], and this showed that the mutant W258X allele was associated with the same haplotype in Korea as in Japan. However, it cannot be concluded that the origin of the W258X allele is common to Japan and Korea because different sets of genetic markers were used in the studies in Japan and Korea.

Immigrants from the Korean peninsula or China helped the recovery of the population in Japan approximately 2300 years ago [29]. Therefore, founder effects are the most likely mechanisms for the spread of the *URAT1* mutant alleles in Japan (Figure 2). The mutant *URAT1* allele R90H is also found in multiple individuals and it is the second most prevalent mutant allele in both Japan and Korea [14,15,16,26]. The expansion of single mutant *URAT1* alleles may be the basis for the high prevalence of hypouricemia in Japan and Korea, but a similar phenomenon has not been identified in other ethnic groups, except in Roma populations in Europe.

Stiburkova et al. reported that the allele frequencies of T467M (1400C > T) and L415-G417 del (1245_1253 del) were 5.56% and 1.92%, respectively, in 1016 Roma who had been randomly selected from five European regions [30]. Both of these mutations differed from those found in Japan and Korea. The authors speculated that the high frequency of *URAT1* mutant alleles in these populations might be explained by the migration of a fairly small group of around 1000 individuals from India approximately 1000 years ago [31,32,33]. Thus, founder effects likely contributed to the high frequency of *URAT1* mutant alleles among the Roma. Although the prevalence of hypouricemia has not been assessed in Roma populations, it may be higher than those in Japan or Korea because the frequencies of the most and the second most prevalent alleles (5.56% and 1.92%, respectively) are both higher than those in Japan (2.37% and 0.40%, respectively).

## 5. Clinical Manifestation of Renal Hypouricemia

EI-AKI and renal stone formation are the two major clinical manifestations of RHUC. Although both were found in ~10% of patients with RHUC who consulted or were referred to clinicians at university hospitals [14], the general prevalence of the symptoms in patients with RHUC has not been published.

Most patients with RHUC and EI-AKI are homozygous or compound heterozygous for mutant *URAT1* or *GLUT9* alleles and exhibit high FEUAs exceeding 30% (it is usually <10% in healthy individuals) [14,15]. Recently, Hosoyamada’s group developed a mouse model of EI-AKI associated with RHUC by knocking out both the *URAT1* and *UOX* (urate oxidase) genes [34,35] and making a genetic modification that increased their hypoxanthine phosphoribosyl transferase activity. After exercising the mice to exhaustion through forced swimming, they showed increases in their serum creatinine concentrations (EI-AKI) and in their urinary excretion of UA, whereas the control mice did not. In addition, the administration of a xanthine oxidoreductase inhibitor (XOI) (allopurinol or topiroxostat) prevented the development of EI-AKI and the high urinary excretion of UA [34,35]. Therefore, the EI-AKI associated with RHUC appears to be the result of renal damage induced by the high concentration of UA in the renal tubules following the exercise.

Yeun et al. showed for the first time that it is possible to reproduce EI-AKI in a patient with RHUC who had experienced episodes of EI-AKI by getting the patient to perform physical exercise to exhaustion in a laboratory [36]. Although it is unknown whether the patient had mutations in their *URAT1* or *GLUT9* genes because the experiment was performed before these genes had been identified, the patient had typical features of RHUC, including a low SUA concentration (0.5 mg/dL) and a high FEUA (55.2–69.4%). After the exercise to exhaustion, the creatinine clearance of the patient decreased (EI-AKI) and their urinary excretion of UA increased. In contrast, neither the creatinine clearance nor the urinary excretion of UA of the four control participants changed following the exercise. Furthermore, the administration of allopurinol at a dose of 300 mg/day for 5 days before the exercise inhibited both the development of EI-AKI and the increase in the urinary excretion of UA in the patient following the exercise. On the basis of these results, the authors concluded that the EI-AKI was induced by an increase in the urinary excretion of UA following the exercise and that the administration of allopurinol could prevent EI-AKI [36] (Figure 3). A further case report showed the effectiveness of allopurinol administration for the prevention of EI-AKI (300 mg for 3 days before a 400 m run) [37].

XOIs, including allopurinol, inhibit the production of oxygen radicals derived from xanthine oxidase [38], and therefore, their effect on preventing EI-AKI does not eliminate another hypothesis that EI-AKI in RUHC patients is the result of vasoconstriction induced by the oxidative stress that develops during exercise to exhaustion, which would be largely prevented by a normal concentration of UA, which is a powerful antioxidant [39]. 

Although studies have not shown whether the prevalence of renal calculi is high in patients with RHUC, greater urinary excretion of UA logically would represent a risk factor for the formation of renal stones because the administration of 300 mg allopurinol was shown to prevent the recurrence of calculi in patients with hyperuricosuria, but not RHUC, in a randomized trial [40].

## 6. Relationships between URAT1 Mutations and Reduced Renal Function

Tabara et al. found that the *URAT1* W258X allele, which is the most common mutant URAT1 allele in Japan, was independently associated with a low estimated glomerular filtration rate (eGFR) after multiple adjustments for clinical factors, including hypertension, SUA concentration, and body mass index (BMI), in two cohorts recruited from different Japanese cities [18]. Wakasugi et al. evaluated the relationship between a low SUA concentration and the prevalence of impaired kidney function (eGFR < 60 mL/min/1.73 m^2^) in a large general population recruited from multiple Japanese cities, and found that the prevalence was higher in men with an SUA concentration ≤ 2.0 mg/dL [8]. Furthermore, Kanda et al. performed a prospective study of the contribution of low SUA to a decline in eGFR in a Japanese cohort [41] and found that low SUA (men, <5.0 mg/dL; women, <3.6 mg/dL) was associated with the decrease in eGFR over time. In a different Japanese cohort study, Mori et al. found that low SUA (≤3.5 mg/dL) was a risk factor for the development of chronic kidney disease (CKD) in women, but not in men [42]. Conversely, Koto et al. found no relationship between renal dysfunction and low SUA concentration [10].

In patients with RHUC, EI-AKI usually resolves within a short period (~2 weeks) with only conservative renal support; however, the long-term effects on the kidney are not known [43]. Low-level pathology may remain and lead to chronic renal dysfunction, especially if recurrent episodes of AKI occur. EI-AKI is usually diagnosed in patients who exhibit overt symptoms, such as vomiting and lower back pain, and who attend a hospital [39,43], and it is unknown whether all patients with impaired kidney function following exercise to exhaustion experience such symptoms.

A potent URAT1 inhibitor, namely, lesinurad, increases the FEUA to >30%, which is a value similar to that of patients with homozygous or compound heterozygous loss-of-function mutations in *URAT1* [44]. This drug causes renal toxicity and reduces creatinine clearance [45], and in some patients, the impaired renal function does not recover until 3 months after the use of the drug is discontinued [45]. The mechanism for the nephrotoxicity of the drug is considered to be the high UA load in the renal tubules because the toxicity is more common in patients with high baseline SUA concentrations who are not taking an XOI [45]. Another potent URAT1 inhibitor, namely, verinurad, also increases the FEUA to a level similar to that of patients with homozygous or compound heterozygous loss-of-function *URAT1* mutations [46] and causes renal dysfunction [47]. These drug studies indicate that a tubular load of UA similar to those of patients with homozygous or compound heterozygous loss-of-function *URAT1* mutations may be capable of inducing renal dysfunction. In contrast, another selective inhibitor of URAT1, namely, dotinurad, does not cause renal toxicity and is now being used clinically in Japan [48]. This drug does increase the FEUA, but only by <10% [49]. Benzbromarone, which has long been used as a uricosuric in Japan, also does not cause renal toxicity, and also increases the FEUA by <10% [50].

## 7. Clinical Significance of Heterozygous States for URAT1 Function

The frequency of mutant *URAT1* alleles is close to 3% in Japan, and therefore, the prevalence of heterozygosity is approximately 6% in the population. In Korea, the frequency of mutant *URAT1* alleles and the prevalence of heterozygosity may be similarly high. In Roma populations, the frequency of mutant URAT1 alleles and the prevalence of heterozygosity is even higher, probably twice those in Japan. Because heterozygous individuals tend to have a higher FEUA than wild-type individuals [11], they may be at higher risk of a high UA load in the renal tubules when the filtration of UA is increased. In fact, EI-AKI was identified in some heterozygous individuals, although almost all of the reported patients with EI-AKI were homozygous or compound heterozygous [23]. Okabayashi et al. reported an impressive case of renal transplantation in which the donor was heterozygous for a *URAT1* mutant allele (a nonsense mutation in exon 5) [51] and the recipient was a son of the donor and also heterozygous for the same *URAT1* mutant allele. There were no clinical complications following the transplantation and a biopsy of the transplanted kidney was performed after 3 months, according to the protocol. Histological analysis of the biopsy showed scattered UA crystals and calcium-based calculi in the distal tubules. The SUA concentrations in the recipient were 5.2 mg/dL and 1.9 mg/dL, before and 3 days after the transplantation, respectively, and his FEUA was 29% after the transplantation. The authors speculated that UA had accumulated in the body of the recipient before the transplantation because of renal insufficiency, but that this was filtered very swiftly and accumulated in the renal tubules after the transplantation [51]. One year later, the UA crystals and calcium-based calculi had disappeared without the use of an XOI. Thus, the kidney tubules of heterozygous individuals are at risk of a high UA load when the filtration of UA is increased.

The identification of heterozygous individuals by means of standard blood and urine tests is rather difficult because the SUA concentration and FEUA of heterozygous individuals are highly variable and overlap with the normal ranges [11,19]. In addition to lifestyle factors, the high prevalence of *ABCG2* mutations, which cause renal UA overload, can affect both the SUA concentration and FEUA of individuals who are heterozygous for *URAT1* [52,53]. Thus, clinicians should bear in mind that heterozygosity for *URAT1* is relatively prevalent in Japan at >5%. Although the prevalence of heterozygosity may be much lower in countries other than Japan and Korea, and in the Roma, it is much higher than in those with homozygosity or compound heterozygosity.

## 8. Conclusions

Patients with RHUC who are homozygous or compound heterozygous for loss-of-function mutations of the *URAT1* gene have SUA concentrations ≤ 1.0 mg/dL and are present at a prevalence of ~0.1% in both men and women. However, many more women than men who are heterozygous for loss-of-function *URAT1* mutations have SUA concentrations of 1.1 to 2.0 mg/dL, and this is the principal contributor to the difference in the prevalence of SUA concentrations ≤ 2.0 mg/dL between the sexes in the general population. The frequency of *URAT1* mutant alleles is nearly 3% in Japan and the spread of a mutant allele from a single individual in ancient times is sufficient to explain this. A similar phenomenon was speculated in Roma populations in which the frequency of a loss-of-function mutant allele is also high.

Impaired reabsorption of filtered UA, resulting in a high UA load on renal tubules following exercise to exhaustion, may be the cause of EI-AKI in patients with RHUC, and this can be prevented by the administration of an XOI. Thus, one of the physiological roles of URAT1 may be the protection of renal tubular cells against the damage induced by a high UA load after exercise to exhaustion, which was presumably more frequent in ancient times, when food was obtained mainly through hunting and gathering. In addition to EI-AKI, chronic renal damage may be caused by mutations in *URAT1*. Because heterozygosity for *URAT1* mutants is much more common than homozygous or compound heterozygous loss-of-function states in the population and is associated with potential risks of kidney damage and renal calculi, clinicians should bear in mind that such individuals are relatively common, especially in Japanese, Korean, and Romani people.

## Figures and Tables

**Figure 1 biomedicines-10-01696-f001:**
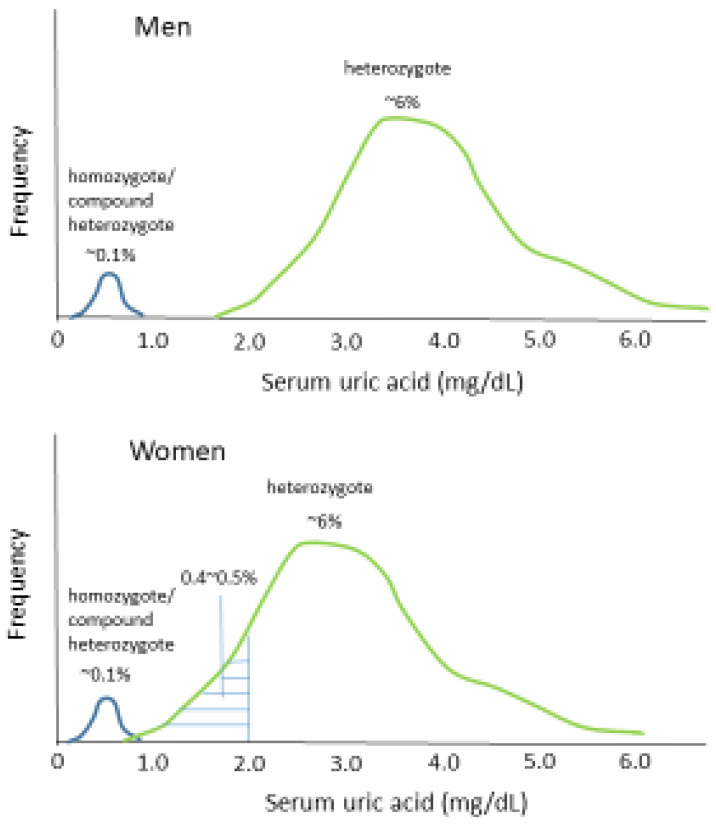
Distribution of serum uric acid concentrations in men and women with a homozygous/compound heterozygous *URAT1* mutation and those with a heterozygous *URAT1* mutation.

**Figure 2 biomedicines-10-01696-f002:**
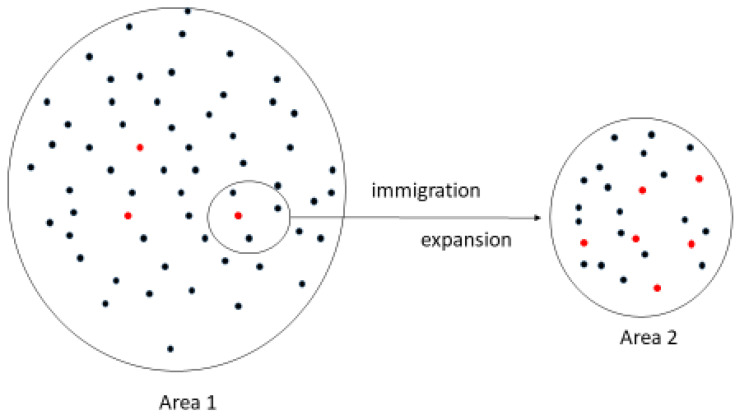
Founder effects. A small number of individuals, including one with *URAT1* mutation (red circle), immigrated to a new area and expanded. As the population expanded in the new area, the number of individuals with the *URAT1* mutation increased.

**Figure 3 biomedicines-10-01696-f003:**
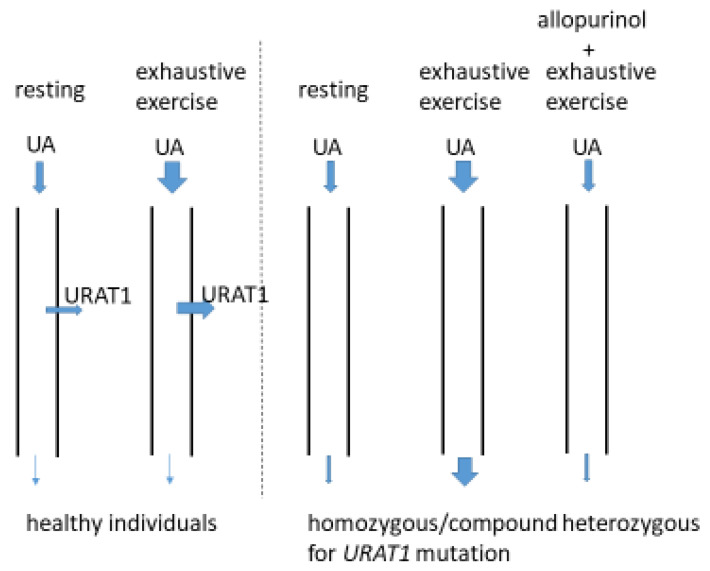
Increased UA load on renal tubules in individuals with homozygous/compound heterozygous *URAT1* mutation after exhaustive exercise. A large amount of UA is produced by exhaustive exercise and then filtered through the glomeruli of the kidneys. In healthy individuals, an increased amount of filtered UA is reabsorbed by URAT1 and the amount of UA excreted into the urine does not increase [36]. On the other hand, in individuals with a homozygous/compound heterozygous *URAT1* mutation, the filtered UA cannot be reabsorbed, resulting in increased excretion of UA in the urine [36]. In this case, it was suggested that the UA load on renal tubules increased after exhaustive exercise. Administration of allopurinol decreased the production of UA by exhaustive exercise and thus inhibited the increase of UA excretion in the urine [36]. EI-AKI did not develop after allopurinol administration [36]. A similar phenomenon will be observed in individuals with homozygous/compound heterozygous *GLUT9* mutation.

**Table 1 biomedicines-10-01696-t001:** Allele frequencies of *URAT1* W258X in Japanese populations.

Reports	Number of Alleles Examined	Number of W258X Alleles	Percentage of W258X Alleles	Area (Japan)
Iwai, N et al. [16]	3750	89	2.37	Suita
Taniguchi, A et al. [17]	1960	45	2.3	Tokyo
Tabara, Y et al. [18]	10330	263	2.55	Suita + Ehime
Hamajima, N et al. [19]	9586	235	2.34	Hamamatsu

**Table 2 biomedicines-10-01696-t002:** Frequency of mutant *URAT1* alleles (W258X and R90H) in individuals with low SUA concentrations [11].

	Men (n = 108)				Women (n = 932)			
SUA (mg/dL)	Number of Deficient URAT1 Alleles				Number of Deficient URAT1 Alleles			
	0	1	2	Total	0	1	2	Total
0.0–1.0	2	4	11	17	0	6	13	19
1.1–2.0	1	3	0	4	20	37	0	57
2.1–3.0	29	58	0	87	570	286	0	856

**Table 3 biomedicines-10-01696-t003:** Prevalence of low SUA concentration in individuals undergoing a general health examination in Japan [11].

	Men (n = 13,607)		Women (n = 17,078)	
SUA (mg/dL)	Number	Frequency (%)	Number	Frequency (%)
0.0–1.0	20	0.15	23	0.13
1.1–2.0	4	0.03	70	0.41
2.1–3.0	107	0.79	1093	6.40

## Data Availability

Not applicable.

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
