# Peer review of "Genetic Basis of the Epidemiological Features and Clinical Significance of Renal Hypouricemia"

_biomedicines, 2022, doi:10.3390/biomedicines10071696_

Round 1

Reviewer 1 Report

The paper deals with a topic of great importance. From the genetic classification point of view, the argument is developed in great detail. Update bibliography. The iconography of the tables is well represented. Fluent scientific english. To implement a paper well done, the authors should better detail the physiopathological mechanisms of hypouricemia, wich intervene in the determinism of renal damage. 

Author Response

Thank you for your valuable and kind comments to our manuscript.

Following your comment, the reference 3 was substituted by a rather newer one that investigated the role of uric acid as an anti-oxidant in humans.

We added in Introduction sentences describing the importance of high uric acid load on renal tubules for the pathogenesis of EI-AKI in patients with RHUC. We also described in Introduction the effects of potent URAT1 inhibitors on the renal function through increased UA load on the tubules.

Reviewer 2 Report

In the article (biomedicines-1763815), the authors made substantial efforts to  to survey the  literature and summarize the latest updates on the genetic Basis of renal hypouricemia. Moreover, a detailed illustration of genetic basis for the epidemiological features of hypouricemia between among different ethnic groups were summarized and have been well described. The authors have also emphasized and illustrated the clinical manifestation of renal hypouricemia, and the correlation between URAT1 mutations and reduced renal function. Overall, the surveyed literature and protocol applied to achieve its purpose are adequate, well-structured, well-presented, and actual. The results provided by the manuscript could be a real gain for researchers in the field. Accordingly, I would recommend the publication of this interesting review article after addressing the following revision and suggestions:

- the authors should modify the abstract to better present the aim of their article and what  they would discuss and cover.

- Also, the introduction is poorly presented and does not provide a suitable intro for the topic.

- I would suggest that the authors include some summarizing figures. This would be definitely informative.

Otherwise, this article is very interesting.

Author Response

Thank you for your valuable and kind comments to our manuscript.

Following your comments, the abstract was revised and a new paragraph describing the contribution of the identification of URAT1 to the understanding of epidemiology and pathophysiology of RHUC was added in the introduction.

Summarizing figures (Fig. 1, 2, 3) were added to the manuscript. Fig. 1 summarized the relationship between the URAT1 genotypes and serum uric acid concentration. Fig. 2 explained the mechanism for the expansion of a specific mutant URAT1 allele in a population (founder effects). Fig. 3 summarized the pathophysiology of EI-AKI in patients with RHUC.